# Identification, Molecular Characterization, and Tissue Expression Profiles of Three *Smad* Genes from Water Buffalo (*Bubalus bubalis*)

**DOI:** 10.3390/genes12101536

**Published:** 2021-09-28

**Authors:** Jie Zhang, Guangle Zhang, Yongwang Miao

**Affiliations:** Faculty of Animal Science and Technology, Yunnan Agricultural University, Kunming 650201, China; zjdnflol@gmail.com (J.Z.); zgl18847551798@163.com (G.Z.)

**Keywords:** *Smads 1*, *4*, and *5*, buffalo, isolation and identification, molecular characteristics, tissue expression profile

## Abstract

Smads are involved in a variety of biological activities by mediating bone morphogenetic protein (BMP) signals. The full-length coding sequences (CDSs) of buffalo *Smads 1*, *4*, and *5* were isolated and identified through RT-PCR in this study. Their lengths are 1398 bp, 1662 bp, and 1398 bp, respectively. In silico analysis showed that their transcriptional region structures, as well as their amino acid sequences, physicochemical characteristics, motifs, conserved domains, and three-dimensional structures of their encoded proteins are highly consistent with their counterparts in the species of Bovidae. The three Smad proteins are all hydrophilic without the signal peptides and transmembrane regions. Each of them has an MH1 domain and an MH2 domain. A nuclear localization sequence was found in the MH1 domain of buffalo *Smads 1* and *5*. Prediction showed that the function of the three Smads is mainly protein binding, and they can interact with BMPs and their receptors. The three genes were expressed in all 10 buffalo tissues assayed, and their expression in the mammary gland, gonad, and spleen was relatively high. The results here indicate that the three buffalo Smads may be involved in the transcriptional regulation of genes in a variety of tissues.

## 1. Introduction

Smad was a family of proteins created with the identification of human *Smad1* according to its sequence similarity to the Sma (small worm phenotype) in *C. elegans* and the Mad (Mothers Against Decapentaplegic) in *Drosophila* [1]. There are eight Smads (*Smads 1–8*) found in vertebrates. Among them, *Smads 1*, *4*, *5*, and *8* serve as signal relay factors in the bone morphogenetic protein (BMPs) pathway, which are involved in a variety of biological activities, including embryonic stem cells (ESCs) self-renewal and organ morphogenesis [2], nervous system development [3], osteoblastogenesis and bone formation [4], gametogenesis [5], folliculogenesis [6], placental angiogenesis [7], early embryogenesis [8,9], and embryo survival [10].

Cattle *Smads 1*, *4*, and *5* are located on the autosomal chromosomes 17, 24, and 7, and consists of 6, 11, and 7 exons, respectively. Their coding sequences (CDSs) are 1398 bp, 1662 bp, and 1398 bp (https://www.ncbi.nlm.nih.gov/gene, accessed on 22 September 2021), respectively. So far, the cDNAs of *Smad1* and *Smad4* have been isolated and identified in cattle [11]. The sequence of cattle *Smad5* has been annotated in the whole genome database of domestic cattle [12]. In cattle, *Smad4* has previously been found to be expressed in the testis, ovary, liver, pancreas, small intestine, thymus gland, skeleton muscle, cardiac muscle, and lymph, suggesting that the *Smad4* is widely expressed [11]. At present, there are few reports on the research of buffalo *Smads*.

The domestic buffalo (*Bubalus bubalis*) plays a key part in the agricultural economy of many subtropical and tropical countries by providing meat, draught power, and milk. It is estimated that the number of buffalo in the world is about 205 million, of which more than 97 percent are distributed in Asia [13]. However, compared to cattle, the reproductive performance and milk production of buffaloes around the world are relatively low, and further improvements are urgently needed [14]. Genetic and gene technologies are an important way to improve buffalo production traits, such as lactation and reproductive traits. Smads serve as signal relay factors in the BMP pathway, which have been found to play a unique role in many important biological processes [4,5,6], but there is less attention on buffalo Smads. In this study, the full-length CDSs of *Smads 1*, *4*, and *5* were isolated from water buffalo, and the analyses of bioinformatics and multi-tissue differential expression were further performed. The results here will lay a foundation for further understanding the biological function of the three genes in water buffalo.

## 2. Materials and Methods

### 2.1. Sample Collection, RNA Extraction, and cDNA Synthesis

The procedures of sample collection were executed according to the Guide for Animal Care and Use of Experimental Animals and allowed by the Institutional Animal Care and Use Committee of Yunnan Agricultural University. Every effort has been made to minimize suffering. Ten healthy adult Binglangjiang buffalo—a kind of river buffalo distributed in the west of Yunnan, China—including five male and five female buffalo, were selected for sample collection in this study. They were all 5–6 years old, and the females were at the peak of milking. They have the same feeding and management conditions and drink clean water freely. After the buffaloes were slaughtered, the liver, heart, lung, spleen, brain, kidney, cerebellum, mammary gland, testis, and ovary were immediately sampled, stored in liquid nitrogen, and then transported back to the laboratory for storage in a −80 °C refrigerator. Total RNA of each tissue was extracted using the RNAiso Plus kit (TaKaRa, Dalian, China) according to product instructions. The RNA was treated with RNase-free DNase I (TaKaRa, Dalian, China) to avoid possible DNA contamination. The integrity of the obtained total RNA was detected with 1.5% agarose gel electrophoresis banded with ethidium bromine. Then, its quality was measured, employing a UV–Vis spectrophotometer. The cDNA was synthesized from the RNA of each sample by M-MLV reverse transcriptase (Invitrogen, Waltham, MA, USA).

### 2.2. Primer Design and Gene Isolation

Three pairs of primers were designed according to the mRNA sequences of cattle *Smad1* (NM_001076223), *Smad4* (NM_001076209), and *Smad5* (NM_001077107) in the NCBI database (http://www.ncbi.nlm.nih.gov/, accessed on 22 September 2021) by Primer Premier 5.0 [15]. Detailed primer information is described in Table 1.

The CDSs of buffalo *Smads 1*, *4*, and *5* were respectively isolated by PCR with a system of 25 μL final volume containing 100 ng cDNA (2 μL), 2.5 μL of 10 × buffer (Mg^2+^ Plus), 0.5 μL for forward and reverse primers, respectively (10 μM), 2.0 μL mixed dNTPs (2.5 mM each) (TaKaRa, Dalian, China), 0.25 μL (5 U μL^−1^) Ex Taq DNA polymerase (TaKaRa, Dalian, China), and 17.25 μL of double-distilled water. The PCR mixture was first denatured at 95 °C for 5 min, followed by 35 cycles, each cycle denaturation at 95 °C for 30 s, annealing with different temperatures depending on different primers, and extension at 72 °C for 2 min, and finally extended at 72 °C for 5 min. The PCR products were detected by the electrophoresis of 1% agarose gel, then purified by a DNA gel extraction kit (Axygen, Hangzhou, China) and ligated to pMD-18T plasmid (TaKaRa, Dalian, China), and further sequenced bi-directionally by an ABI PRISM 3730 DNA sequencer (ABI, Foster City, CA, USA) following the manufacturer’s manual. At least 10 clones were sequenced.

### 2.3. Analysis of Gene Sequence and Structure

The open reading frame (ORF) was predicted by the Open Reading Frame Finder program (https://www.ncbi.nlm.nih.gov/orffinder/, accessed on 22 September 2021), and the amino acid sequences (AASs) were deduced from the CDSs of buffalo *Smads 1*, *4*, and *5* via EditSeq program of Lasergene 7 software package (DNAStar, Inc., Madison, WI, USA). Codon usage bias analysis was performed by codonW (http://codonw.sourceforge.net/, accessed on 22 September 2021). The genetic structure analysis of buffalo and its comparison with other Bovidae species are as follows: The transcript information of *Smads 1*, *4*, and *5* of buffalo and other Bovidae species were obtained from the GTF (General Transfer Format) files which downloaded from NCBI datasets (https://www.ncbi.nlm.nih.gov/datasets/, accessed on 22 September 2021). Then, the mRNA and UTR information was added to these GTF files by the GXF fix function of TBtools software [16]. After calculating the relative position of each transcript and each exon in Excel, the information was summarized into a text file in the BED (Browser Extensible Data) format, and then submited to the Gene Structure Display Server 2.0 (http://gsds.gao-lab.org/, accessed on 22 September 2021) [17] for visualization of the gene structures, including untranslated regions, exons, and introns.

### 2.4. Physiochemical Characterization, Motif and Structure

Physicochemical characteristics, including hydropathy, signal peptide, transmembrane region, theoretical molecular weight (MW), and isoelectric point (PI) were predicted using ProtScale (http://web.expasy.org/protscale/, accessed on 22 September 2021), SignalP 5.0 Server (http://www.cbs.dtu.dk/services/SignalP/, accessed on 22 September 2021) [18], TMHMM version 2.0 (http://www.cbs.dtu.dk/services/TMHMM/, accessed on 22 September 2021) [19], the ProtParam tool (http://web.expasy.org/protparam/, accessed on 22 September 2021), and Compute pI/Mw tool (http://web.expasy.org/compute_pi/, accessed on 22 September 2021) [20], respectively. Clustal X 2.0 is used for multiple sequence alignment [21]. The identity and divergence of amino acid sequences were built by MegaAlign program of Lasergene 7 software package (DNAStar, Inc., USA), and further visualized by MORPHEUS (https://software.broadinstitute.org/morpheus, accessed on 22 September 2021). Phylogenetic trees based on the amino acid sequences of *Smad1/4/5* were constructed by the maximum likelihood method (the JTT+G matrix model) using MEGA7 [22], and then the results of the construction tree were exported. The sequences of *Smads 1*, *4*, and *5* in each species were submitted to the MEME website (http://meme-suite.org/tools/meme, accessed on 22 September 2021) for conserved motif confirmation. The conserved domain analysis was performed by submitting the *Smad 1*, *4*, and *5* sequences of all species to NCBI Batch Web CD-Serach tool (https://www.ncbi.nlm.nih.gov/Structure/bwrpsb/bwrpsb.cgi, accessed on 22 September 2021) [23]. Finally, the results of the above three analyses were input into the Gene Structure View (Advance) tool of TBtools to visualize the final merge results [16].

Secondary structures of the deduced AASs were predicted by SOPMA (http://npsa-pbil.ibcp.fr/, accessed on 22 September 2021). The tertiary structures of the *1*, *4*, and *5 Smads* were predicted using the SWISS-MODEL (http://swissmodel.expasy.org/, accessed on 22 September 2021) [24] with a homologous modeling method. The nuclear localization sequence (NLS) prediction was conducted by cNLS mapping (http://nls-mapper.iab.keio.ac.jp/cgi-bin/NLS_Mapper_ref.cgi, accessed on 22 September 2021) [25,26,27]. cNLS Mapper extracts putative NLS sequences with a score equal to or more than the selected cut-off score. A GUS-GFP reporter protein fused to an NLS with a score of 8, 9, or 10, is exclusively localized to the nucleus, that with a score of 7 or 8 partially localized to the nucleus, that with a score of 3, 4, or 5 localized to both the nucleus and the cytoplasm, and that with a score of 1 or 2 localized to the cytoplasm. The subcellular localizations were performed by Euk-mPLoc 2.0 (http://www.csbio.sjtu.edu.cn/bioinf/euk-multi-2/, accessed on 22 September 2021) [28]. Amino acid modifications were predicted by Prosite Scan (http://prosite.expasy.org/prosite.html, accessed on 15 November 2020). SIRING (https://string-db.org/, accessed on 22 September 2021) was used to predict protein-protein interactions. Protein functional analysis was performed by InterProScan (http://www.ebi.ac.uk/interpro/search/sequence-search, accessed on 22 September 2021).

### 2.5. RT-qPCR and Tissue Differential Expression

In this study, according to the obtained CDSs of buffalo *Smads 1*, *4*, and *5*, three pairs of primers were designed to detect their tissue differential expression. The relative expression levels of *Smads 1*, *4*, and *5*, in ten tissues, were assayed by RT-qPCR using SYBR Premix Ex Taq (Takara, Dalian, China) and conducted on iQ5 Real-Time PCR (Bio-Rad, Hercules, CA, USA) followed the manufacturer’s instructions. Each reaction mixture contained 2 μL cDNA, and 0.5 μL or 10 μM for forward and reverse primers, respectively, 7 μL double-distilled water, and 10 μL SYBR Premix Ex Taq. The qPCR was executed initially at 95 °C for 30 s, then followed by 35 cycles with each cycle at 95 °C for 5 s, at 60 °C for 20 s and at 72 °C for 30 s. The *β*-actin (*ACTB*; NM_001290932), ribosomal protein S15 (*RPS15*; XM_006050525) and ribosomal protein S23 (*RPS23*; XM_006059350) were used as endogenous references for normalization of the *Smads 1*, *4*, and *5* expression levels Table 1. The qPCR data were analyzed using the method of −2${}^{\Delta \Delta \mathrm{Ct}}$. The ΔCt = Ct ${}_{\left(target\;gene\right)}$ − Ct ${}_{\left(geometric\;mean\;of\;three\;reference\;genes\right)}$. ΔΔCt = ΔCt − ΔCt${}_{\left(median\right)}$. The geometric mean of the Ct values of the three reference genes obtained in this study was used to normalize the targeted mRNA expression.

## 3. Results

### 3.1. Isolation and Identification of Buffalo Smad1/4/5

Using the mixed cDNA as the templates, the full-length CDSs of buffalo *Smads 1*, *4*, and *5* were isolated by PCR. The PCR products obtained were 1542 bp, 1953 bp, and 1612 bp in length, respectively (Figure 1). They, in turn, contain an open reading frame of 1398 bp, 1662 bp, and 1398 bp. The homology search showed that the CDSs from the three amplicons of 1542 bp, 1953 bp, and 1612 bp had high similarity with those of *Smads 1*, *4*, and *5* of other common species of Bovidae (more than 97.85%). Therefore, the obtained sequences here are the sequences of buffalo *Smads 1*, *4*, and *5*. Then the CDSs of buffalo *Smads 1*, *4*, and *5* were submitted to the NCBI database under the accession numbers KF472050, KF472051, and KF472052.

### 3.2. Gene Sequence and Structure Analysis

The CDSs of buffalo *Smads 1*, *4*, and *5* and their amino acid sequences are presented in in Figure 2.

The results of codon usage bias analysis for buffalo *Smads 1*, *4*, and *5* are shown in Appendix A. The ENc (effective number of codons), contents of GC, and GC3s (the guanine or cytosine frequency at the third codon position of a synonymous codon, excluding methionine, tryptophan, and stop codons) for buffalo *Smads 1*, *4*, and *5* in this study are closer to those of other species in Bovidae. The RSCUs (relative synonymous codon usage) of buffalo *Smads 1*, *4*, and *5* are highly similar to those of cattle. The top three RSCU values for the buffalo *Smad1* are CUG (2.33), AGC (2.08), and UGC (1.86), those for the buffalo *Smad4* are AGU (2.05), GGA (2.05), and UUU (1.85), and for the buffalo *Smad5* are AUU (2.12), AGC (1.80), and GCA (1.80). Codon CUA and AUA are not used in buffalo *Smad1*, and codon GCG is not used in buffalo *Smad5*, but there are no unused codons in buffalo *Smad4*.

The gene structures of buffalo *Smads 1*, *4*, and *5* and their comparison with those of six other species of Bovidae are showed in Figure 3 and Appendix A. For the *Smads 1*, *4*, and *5*, the length of CDS for each gene was almost the same across species of Bovidae. However, the lengths of the UTR (5′UTR and 3′UTR) and introns in them are inconsistent among different species (Figure 3). Among the species of Bovidae, the *Smad1* gene commonly contains several alternative splicing transcripts, while the *Smad4* and *Smad5* contain a small number of the transcripts. In buffalo, *Smad1* contains five kinds of alternative splicing transcripts with inconsistent 5’UTR, one of which has a codon deletion in exon 4 (this was also found in sheep), but only one alternative splicing transcript was found for *Smad4* and *Smad5* in buffalo. It is worth noting that the three genes can be located on different DNA strands in the same species. For example, *Smads 1* and *5* are located on the positive strand of DNA in *Bubalus bubalis*, *Bison bison bison*, and *Capra hircus*, while *Smad4* are located in the reverse strand of DNA in *Bubalus bubalis* and *Bison bison bison*.

### 3.3. Basic Physicochemical Properties of the Three Proteins

There were no significant differences in the basic physicochemical characteristics of *Smads 1*, *4*, and *5* between buffalo and cattle (*Smad1*, NM_001076223.2; *Smad4*, NM_001076209.1; *Smad5*, NM_001077107.3) (Table 2). The *Smads 1*, *4*, and *5* are all unstable proteins without transmembrane regions and signal peptides for both buffalo and cattle. According to the values of GRAVY, the *Smads 1*, *4*, and *5* of buffalo and cattle are all hydrophilic (Table 2, Appendix A).

### 3.4. Sequence Identities and Phylogenetic Analysis

The alignment of amino acid sequences showed that buffalo *Smads 1*, *4*, and *5* have high consistency with those of other species previously published in the NCBI database, especially with those of the species in Bovidae (Figure 4). The sequences of buffalo *Smads 1*, *4*, and *5* all have more than 99.4% similarity with its homologous sequences of other species in the family Bovidae and more than 98.5% similarity with other mammalian species (Figure 4).

In the phylogenetic trees based on the amino acid of 15 species in mammals, *Smads 1*, *4*, and *5* were clustered in their own clade, and the genetic relationship between buffalo and other species of Bovidae is relatively close (Figure 5).

To explore the common structural characteristics and functional similarities of the *Smad 1*, *4*, and *5* proteins, the analyses of the motif patten and conserved domain were taken based on their amino acid sequences in the above species (Figure 6). In 15 mammalian species, nine to ten conserved motifs were found in the *Smads 1*, *4*, and *5* (Figure 6B). Among 15 mammalian species, we found that similar motif patterns exist in *Smad4*, and in both of *Smad1* and *Smad5*. The two ends of *Smad4* had similar motifs to the two ends of *Smad1* and *Smad5*, but the motifs in the middle of *Smad4* and *Smad1/5* were different (Figure 6B). Motif 9 only exists in *Smad1* and *Smad5*, and motif 10 only exists in *Smad4*. For motif 6, there is only one copy in both *Smad1* and *Smad5*, but for *Smad4*, only *Sus scrofa* and *Bos taurus* have one copy, and all the other 13 species have two copies (Figure 6B). More detail information about these motifs in the three Smads, such as E-value, sites, and window width is presented in Table 3.

The prediction showed that buffalo *Smads 1*, *4*, and *5* all contain two conserved functional domains coupled at a link region, which are the same as in other species (Figure 6C): a N-terminal Mad Homology 1 domain (MH1, aa 9–132 for the *Smad1*, aa 14–138 for the *Smad4*, aa 10–133 for the *Smad5*), which belongs to MH1 superfamily, and a C-terminal Mad Homology 2 domain (MH2, aa 265–465 for the *Smad1*, aa 321–542 for the *Smad4*, aa 265–465 for the Smad5), which belongs to MH2 superfamily. In buffalo, *Smads 1* and *5*, a Ser-Ser-X-Ser (SSVS) motif was found at the end of the MH2 domain. In addition, there is a PRK10263 domain which belongs to PRK10263 superfamily in the *Smad1*, and a Pneumovirinae attachment membrane glycoprotein G domain which belongs to Pneumo_att_G superfamily in the *Smad4*.

### 3.5. Structure of Buffalo Smads 1, 4, and 5

Buffalo *Smad1* consists of 22.58% α-helix (105 AAs), 17.85% extended strand (83 AAs), 5.38% β-turn (25 AAs), and 54.19% random coil (252 AAs); buffalo *Smad4* contains 24.41% α-helix (135 AAs), 15.55% extended strand (86 AAs), 6.69% β-turn (37 AAs), and 53.35% random coil (295 AAs), and buffalo *Smad5* is composed of 22.15% α-helix (103 AAs), 19.35% extended strand (90 AAs), 5.16% β-turn (24 AAs), and 53.33% random coil (248 AAs) (Appendix A).

Three-dimensional structures of buffalo *Smad1/4/5* were built based on the method of homologous modeling.The sequence consistencies of buffalo *Smads 1*, *4*, and *5* with their human counterparts (1khu. 1.a, 1DD1.b, and 1Khu. 1.b) were 99.54%, 100%, and 93.09%, respectively, and the coverage rates were 47%, 48%, and 47%, respectively. Three-dimensional structures of the three buffalo Smads are highly similar to those of cattle (Appendix A).

### 3.6. Subcellular Location and Molecular Function

NLS prediction showed that buffalo *Smad1* contained an NLS in aa 12–42 with a score of 5.4, while the Smad5 contained an NLS in aa 13–43 with a score of 5.4. However, NLS was not found in the Smad4. The NLSs of *Smads 1* and *5* are located in their MH1 domain. The cut-off score of 5.4 indicates that the *Smads 1* and *5* are located in both nucleus and cytoplasm. Subcellular localization prediction also showed that buffalo *Smads 1*, *4*, and *5* play roles in the cytoplasm and nucleus. The prediction showed that the *Smads 1*, *4*, and *5* proteins are involved in the biological processes related to DNA-templated gene transcription regulation (GO:0006355) and TGF-β (transforming growth factor beta) receptor signaling pathway (GO:0007179). Their molecular function is mainly protein binding (GO:0005515), and their cellular component is the transcription regulator complex (GO:0005667).

The protein-protein interaction network was constructed by using the program SIRING (Appendix A). The three Smad proteins were found to interact with 17 other proteins, which included some TGF-β family receptors (such as BMPR1A, ACVR2A), SKI, and other Smads. In addition, the *Smad4* and *Smad5* can interact with a ubiquitin-protein ligase (SMURF2 or NEDD4L), the *Smad1* and *Smad5* can interact with ZFYVE9, the *Smad1* can interact with YAP1, and the *Smad4* can interact with BMP2 and SKIL.

### 3.7. Amino Acid Modifications

The prediction of modification sites revealed that there were five, five, and four types of amino acid modification sites in the *Smad1*, *Smad4*, and *Smad5*, respectively (Figure 7), including N-glycosylation site, N-myristoylation site, protein kinase C phosphorylation site, and casein kinase II phosphorylate site (Appendix A).

### 3.8. Tissue Expression Profile

The expressions of *Smads 1*, *4*, and *5* genes were detected in 10 buffalo tissues in this study (Figure 8). For the male and female buffalo, the expression levels of *Smads 1*, *4*, and *5* in most of the assayed tissues were inconsistent. For example, in male buffalo, the relative expression levels of *Smad4* and *Smad5* in the heart and spleen, *Smad5* in the liver and kidney, *Smads 1*, *4*, and *5* in the lung, brain, and cerebellum were significantly higher than those in female buffalo (*p* < 0.01), and the expression levels of *Smad1* and *Smad4* in the liver and kidney of male buffalo were significantly lower than those of female buffalo (*p* < 0.01). It was worth noting that the expression abundance of the three genes in the mammary gland, ovary, and testis was relatively high in all the tissues tested.

## 4. Discussion

A total of eight Smads are found in mammals, which are divided into three distinct subtypes: the first category includes *Smads 1*, *2*, *3*, *5*, and *8*, which are receptor-regulated Smads (R-Smads). The second type acts as common partner Smad (Co-Smad) for all R-Smads, including only Smad4. The third category is inhibitory Smads (I-Smads), including *Smads 6* and *7* [29]. In this study, the CDS of *Smads 1*, *4*, and *5* was isolated and identified from water buffalo. Only one type of CDS was identified in each of the three buffalo *Smads*. The length, base composition, and RSCU value of the CDS of buffalo *Smads 1*, *4*, and *5* are highly consistent with that of their counterparts in other Bovidae species. In addition, the structure of the transcriptional regions of the three *Smad* genes of buffalo is also highly similar to that of other Bovidae animals. The physicochemical characteristics, motifs, conserved domains, and three-dimensional structures of buffalo *Smads 1*, *4*, and *5* are also highly consistent with those of other mammalian species. The alignment of amino acid sequences showed that buffalo *Smads 1*, *4*, and *5* have high consistency with those of other mammals. Phylogenetic analysis also displayed that the three buffalo Smads were clustered in their own clade, each of which showed that the buffalo has a close genetic relationship with other mammals. The above results indicate that buffalo *Smads 1*, *4*, and *5* have similar functions with those of other mammals, especially the species in Bovidae, and that the three genes in mammals are functionally conserved.

*Smads 1*, *4*, and *5*, as signal transduction molecules, are an important part of the mammalian BMP signaling pathway, which can transmit signals from membrane receptors to the nucleus [30]. As members of the TGF-*β* superfamily, BMPs constitute a highly conservative multifunctional growth factor family. They have various functions, such as neurogenesis, skeletal formation, hematopoiesis, and embryogenesis [31]. Previous studies have shown that BMP-mediated receptor phosphorylation can initiate the BMP signaling [32]. After BMPs bind to the type II receptor, the type I receptor is phosphorylated, then it further phosphorylates the conserved C-terminal Ser-Ser-X-Ser (SSXS) motif of R-Smads (*Smads 1*, *5* and *8* in BMP pathway). The phosphorylated R-Smads further combine with Smad4 to form a transcriptional complex, which enters the nucleus, where the complex either binds to the specific enhancer/promoter sequence of the target gene alone, or cooperates with some DNA binding subunits to simultaneously activate or repress hundreds of target genes under tightly controlled conditions [33]. In this study, the conserved C-terminal SSVS motif has also been found in buffalo *Smads 1* and *5*, which also support the theory that buffalo *Smads 1* and *5* can be phosphorylated by membrane receptors. Bioinformatic predictions indicate that the main functions of buffalo *Smads 1*, *4*, and *5* are to interact with many proteins in the BMP signaling pathway, and ultimately form a transcriptional regulatory complex that participates in the regulation of target genes, suggesting that the three Smads probably play pivotal roles in buffalo BMP pathway.

The Smad complex mentioned above has a weak affinity for DNA, and its specificity is limited [30]. How they accurately regulate hundreds of BMP signal target genes in the same cell is the focus of current research. One mainstream view is that the key to precise control lies in the existence of post-translational modifications at several functional sites of Smad proteins [34]. Based on the function sites, the Smad complex can cooperate with other site-specific transcription factors and a variety of coactivators and corepressors to modify histones, modulate chromatin structure, stabilize their DNA binding, and further activate or repress target genes directly or indirectly [30]. In this study, some modification sites were predicted in each of the three buffalo proteins. It is speculated that they are closely related to the precise regulation of intracellular BMP signaling. Whether and how these function sites are involved in the precise control of buffalo Smads needs further study.

Previous studies have shown that Smad proteins mediate the signals of BMPs, which are involved in a variety of biological activities. It has been found that the BMP-Smad signaling pathway is involved in heart development [35], embryonic cerebellum development [36], and nervous system development [3]. During early embryonic development and reproductive system development, the genes involved in the BMP signaling pathway have been found to affect the early regulatory processes of porcine oogenesis [37]; in cattle and buffalo, BMP and its receptors have been found to regulate the process of follicle development through granulosa cell steroidogenesis [32,38], and are stage-specifically expressed in the corpus luteum of buffalo [39]. In addition, BMP signaling can induce specific primordial germ cells, which are further recruited to participate in several other germline-specific and somatic reproductive roles. In this study, we found that Smads are highly conserved in structure and function. From this, we infer that the three Smads are also involved in a variety of biological processes of buffalo.

Studies in cattle and Hu sheep have shown that Smads, including *Smads 1*, *4*, and *5* are expressed in various tissues as the transcription factors in the BMP-Smad pathway [40,41]. In this study, the tissue differential expression analysis showed that the *Smads 1*, *4*, and *5* were expressed in all male and female buffalo tissues assayed, indicating that they play a role in a variety of tissues. Furthermore, some research has suggested that the TGF-*β*/BMP pathway may be involved in immune response, and the spleen plays important roles in the immune system [42]. In this study, the *Smads 1*, *4*, and *5* are highly expressed in the spleen of male and female buffalo, suggesting that the three buffalo *Smads* may also participate in the immune response. It is worth noting that buffalo *Smads 1*, *4*, and *5* were relatively highly expressed in the mammary glands and gonads, which means that they may play an important role in these tissues, and they probably participate in the reproductive process of buffalo.

## 5. Conclusions

In this study, buffalo *Smads 1*, *4*, and *5* were isolated and characterized. The physicochemical properties, motifs, conservative domains, and secondary and three-dimensional structures of buffalo *Smads 1*, *4*, and *5* are highly similar to those of *Smads 1*, *4*, and *5* in cattle. Furthermore, the three proteins are functionally conserved in mammals. Buffalo *Smads 1*, *4*, and *5* can interact with BMPs and their receptors, and they probably serve as signal transducers to relay signals from membrane receptors to the nucleus in BMP signaling pathway. The three buffalo *Smads* were expressed in all examined tissues, especially with relatively high mRNA abundance in the mammary gland, gonad, and spleen. The results here suggests that buffalo *Smads 1*, *4*, and *5* may play key roles in the regulation of gene transcription in a variety of tissues, and these roles may be closely related to the lactation, reproduction, and immune response processes of buffalo. This study will provide a basis for elucidating the genetic mechanism of buffalo production traits and high disease resistance.

## Figures and Tables

**Figure 1 genes-12-01536-f001:**
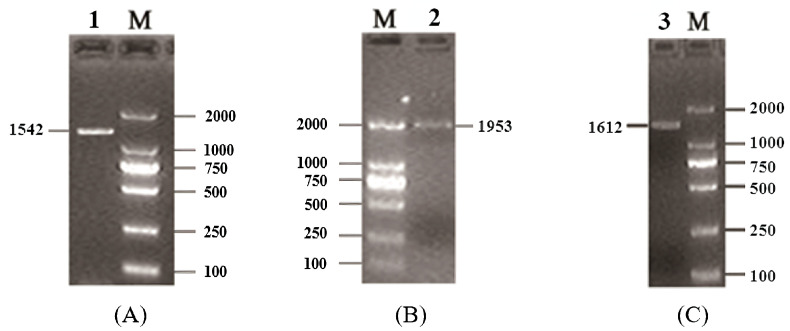
RT-PCR results for buffalo *Smad1* (**A**), *Smad4* (**B**), and *Smad5* (**C**) genes. Lane 1, 2, and 3 are PCR products for the buffalo *Smads 1*, *4*, and *5* genes, respectively. Lane M, DL2000 DNA marker.

**Figure 2 genes-12-01536-f002:**
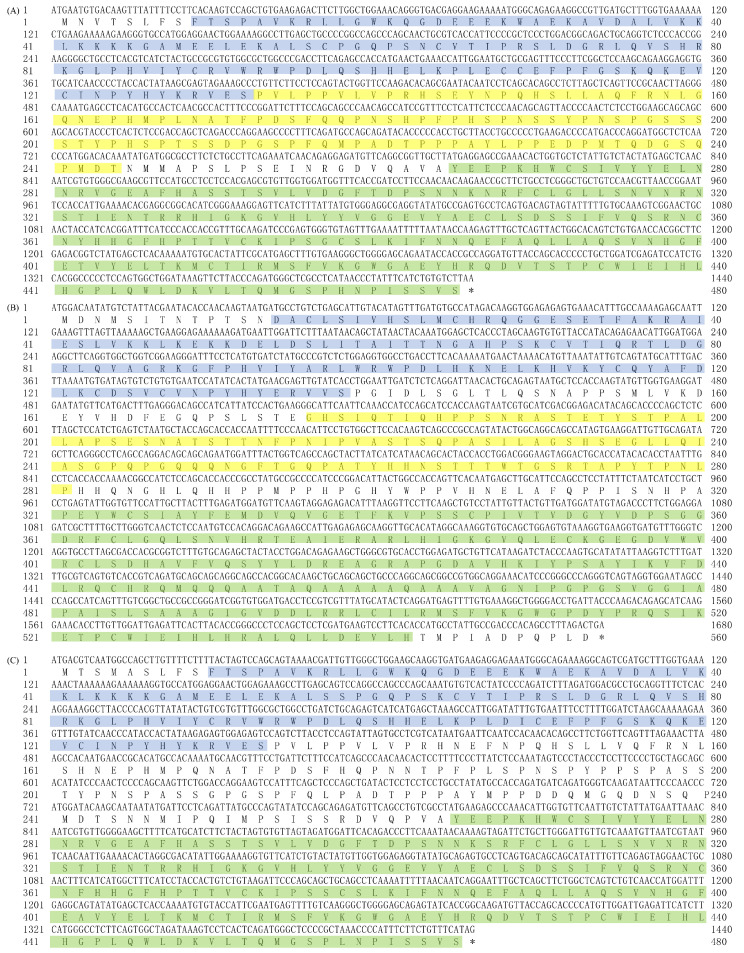
Complete CDSs of buffalo *Smads 1*, *4*, and *5* genes and AASs. (**A**–**C**) exhibit buffalo *Smad1*, *4*, and *5* genes, respectively. The shaded parts of blue/yellow/green represent putative conserved functional domains of MH1_SMAD_1_5_9, PRK10263, and MH2_SMAD_1_5_9 in *Smad1*, and those of MH1_SMAD_4, Pneumo_att_G, and MH1_SMAD_4 in *Smad4*, respectively. The shaded parts in blue/green represent putative conserved functional domains of MH1_SMAD_1_5_9 and MH2_SMAD_1_5_9 in *Smad5*, respectively. “*”, termination codon.

**Figure 3 genes-12-01536-f003:**
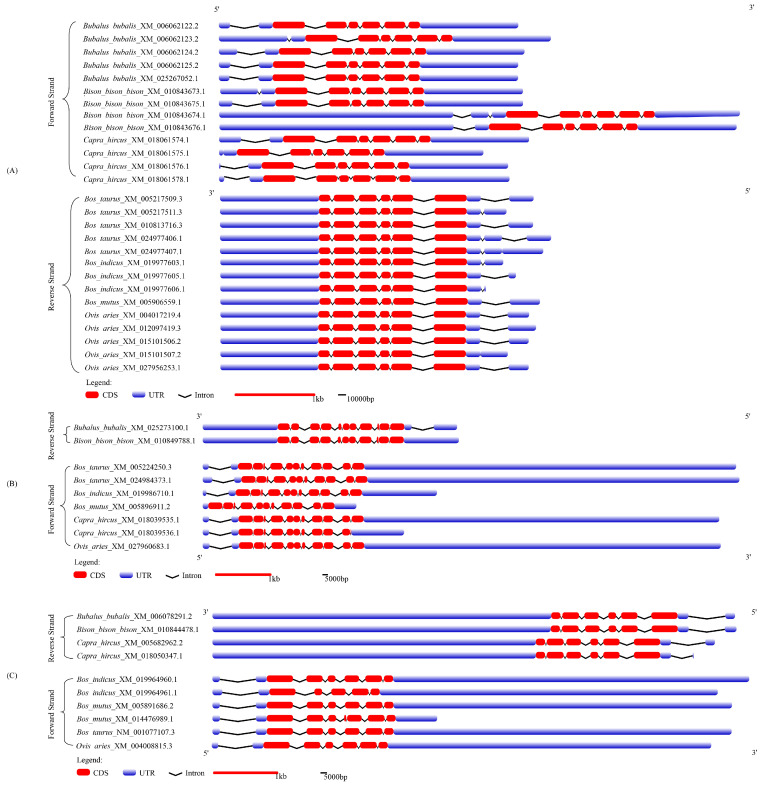
The gene structure of *Smads 1*, *4*, and *5*. (**A**) *Smad1*, (**B**) *Smad4*, (**C**) *Smad5*. The red boxes, blue boxes, and the black hat lines represented untranslated regions (UTRs), CDSs, and introns. The length of the exon and intron can be inferred from the legend at the bottom.

**Figure 4 genes-12-01536-f004:**
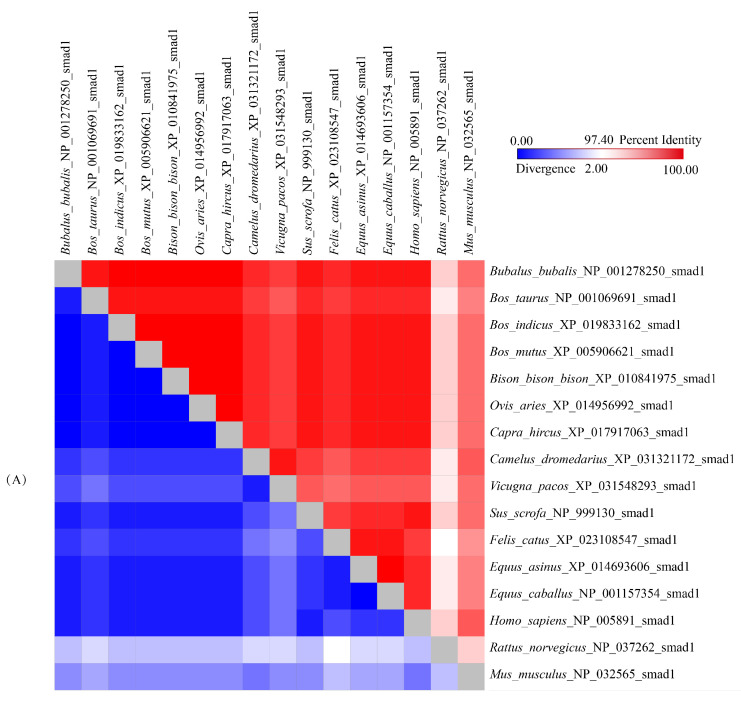
Divergences and percent identities according to the AASs of *Smads 1*, *4*, and *5* between buffalo and other species. (**A**) *Smad1*, (**B**) *Smad4*, and (**C**) *Smad5*. The divergences and percent identities were shown in the upper and lower triangle of the matrix, respectively.

**Figure 5 genes-12-01536-f005:**
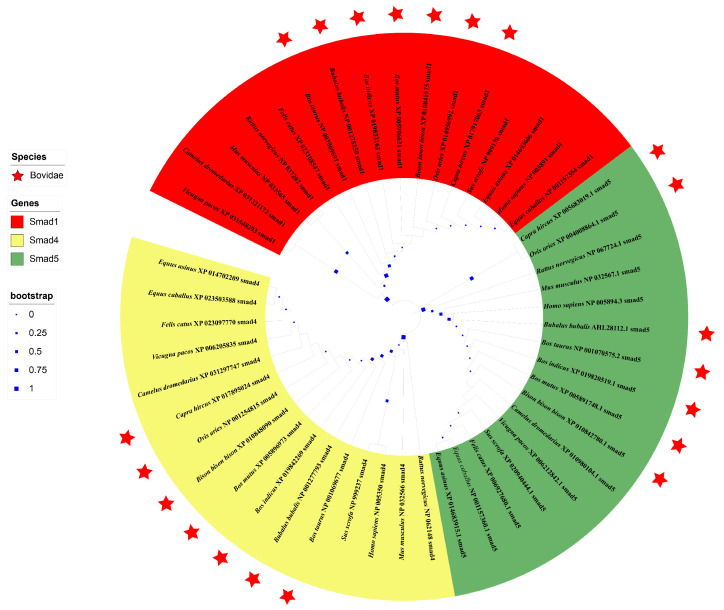
Phylogenetic trees constructed based on the AASs of *Smads 1*, *4*, and *5* in buffalo and other mammals. The bootstrap values of each branch in the tree were labelled with blue boxes of different sizes. The species in the family of Bovidae were labelled by a red star.

**Figure 6 genes-12-01536-f006:**
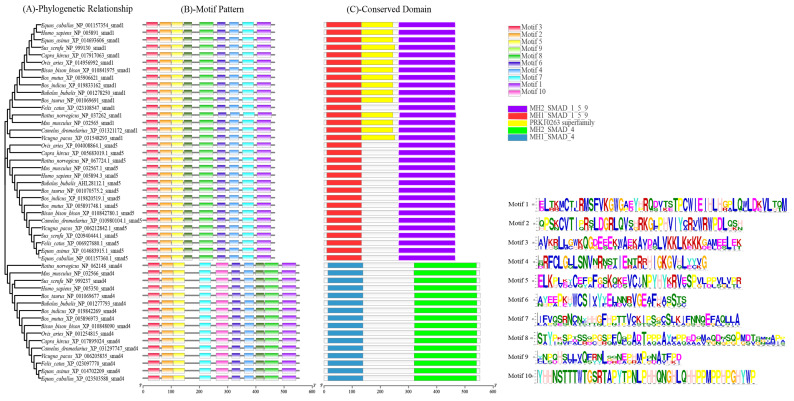
The phylogenetic relationships, motif patterns and conserved domains of *Smads 1*, *4*, and *5* in 15 mammalian species. (**A**) Phylogenetic relationships; (**B**) motif pattern; (**C**) conserved domains. Ten conserved motifs and five kinds of conserved domains in *Smads 1*, *4*, and *5* are labelled with different color boxes.

**Figure 7 genes-12-01536-f007:**
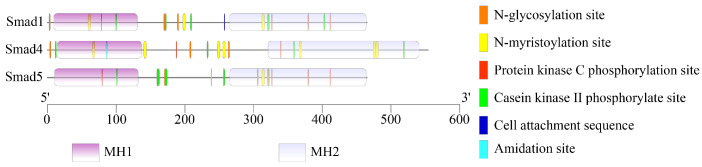
The putative modification sites in buffalo *Smads 1*, *4*, and *5*.

**Figure 8 genes-12-01536-f008:**
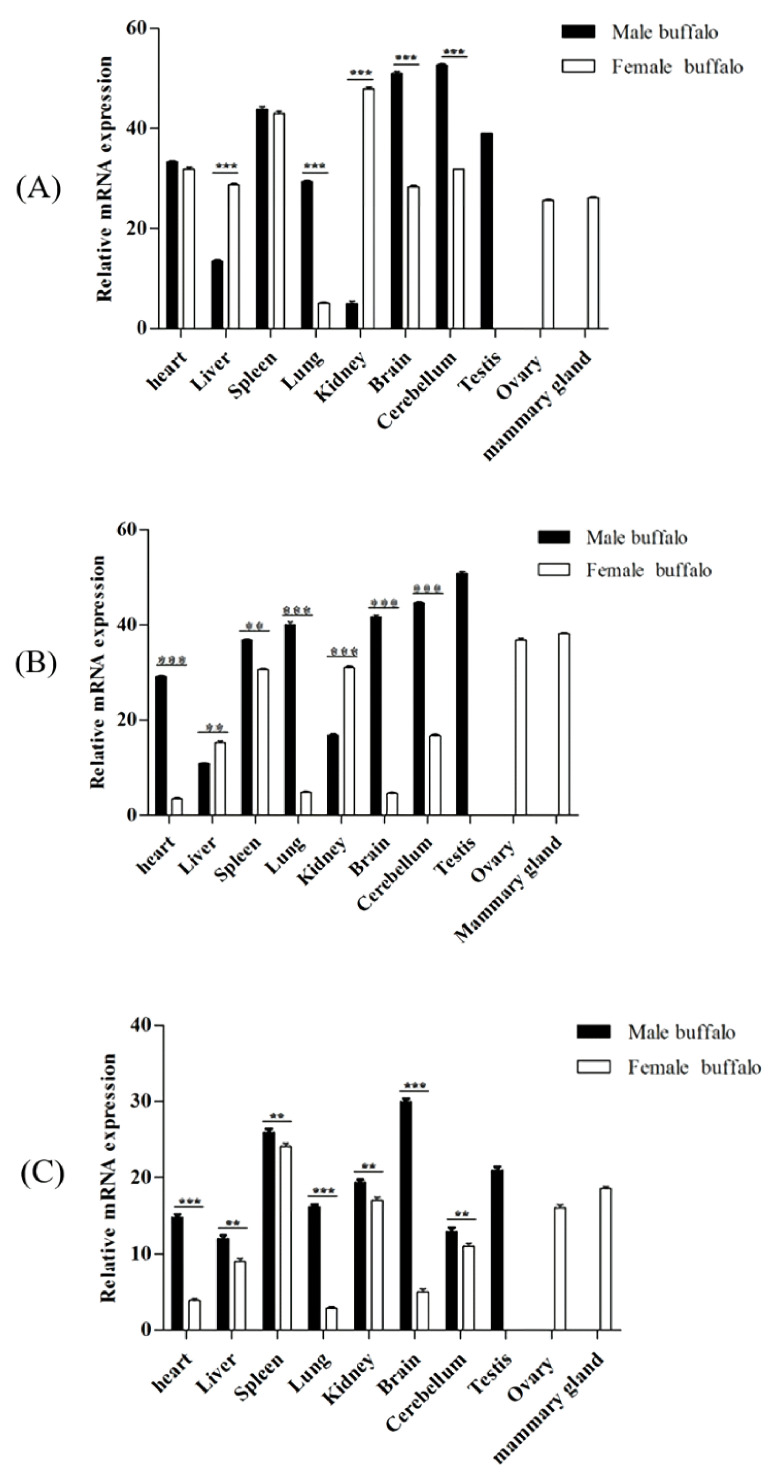
Multi-tissue expression profiles of buffalo *Smads 1*, *4*, and *5* genes in male and female buffalo. (**A**) *Smad1*, (**B**) *Smad4*, (**C**) *Smad5*. The values are provided as means ± SE (** *p* < 0.01, *** *p* < 0.001).

**Table 1 genes-12-01536-t001:** Primers used for isolating buffalo *Smad1*, *Smad4*, and *Smad5* genes.

Gene	Primers(5′–3′)	Amplicon Length (bp)	Annealing Temperature (°C)	Purpose
*Smad1*	F: AAGCCTCTTTCACTGTCCTTTCAC	1542	60	CDS isolation
	R: GTCTGACTCGTCCATCCTTCAAGT			
*Smad4*	F: GCTGCCTCCGAAAGATCAAAAT	1953	55	CDS isolation
	R: GCTCTGAGCCATGCCTGAAAAGT			
*Smad5*	F: TGGCACTTATGAAGATAAAGGTCA	1612	55	CDS isolation
	R: CTGTACTGGAAGTTTCCCTAAAAT			
*Smad1*	F: CCCTCACTCTCCGACCAG	238	57.1	Differential expression
	R: ACCATCCACCAACACGCT			
*Smad4*	F: TCATTATCCACTGAAGGGCATT	173	54	Differential expression
	R: ATACTGGCGGGCTGACTTGT			
*Smad5*	F: TTTCCCTTATCTCCAAAT	286	51.9	Differential expression
	R: CGACAGGCTGAACATCTC			
*ACTB*	F: TGGGCATGGAATCCTG	196	57.9	internal reference
	R: GGCGCGATGATCTTGAT			
*RPS23*	R: CAGGAAGTGTGTCAGGGT	154	50.8	internal reference
	F: TCCAGGAATGTCACCAAC			
*RPS15*	R: GATGGTGGGCAGCATGGTT	159	58.6	internal reference
	F: AAGCGGGAGGAATGGGTGG			

**Table 2 genes-12-01536-t002:** Basic characteristics of *Smad1/4/5* for buffalo and cattle.

Items	*Smad1*		*Smad4*		*Smad5*	
	Buffalo	Cattle	Buffalo	Cattle	Buffalo	Cattle
Isoelectric point (PI)	6.9	6.9	6.5	6.5	7.63	7.63
Molecular weight (MW)	52.25 kDa	52.22 kDa	60.57 kDa	60.54 kDa	52.33 kDa	52.25 kDa
Formula	$C_{2318}H_{3537}N_{645}O_{688}S_{25}$	$C_{2317}H_{3535}N_{645}O_{687}S_{25}$	$C_{2675}H_{4140}N_{764}O_{802}S_{23}$	$C_{2674}H_{4138}N_{764}O_{801}S_{23}$	$C_{2334}H_{3569}N_{643}O_{683}S_{24}$	$C_{2328}H_{3565}N_{643}O_{683}S_{24}$
Strongly acidic amino acids (E, D)	44	44	51	51	42	42
Strongly basic amino acids (R, K)	42	42	44	44	43	43
Polar amino acids (S, T, Y, N, C, Q)	148	147	160	159	145	145
Hydrophobic amino acids (F, W, V, A, I, L)	126	126	173	173	132	132
Instability index (II)	60.84	60.24	50.95	50.83	64.01	64.42
Grand average of hydropathicity (GRAVY)	−0.568	−0.567	−0.389	−0.388	−0.517	−0.516
Aliphatic index (AI)	65.35	65.35	76.06	76.06	68.09	68.09

**Table 3 genes-12-01536-t003:** E-value, sites, and widths of *Smad1/4/5* conserved motifs.

	Motif1	Motif2	Motif3	Motif4	Motif5	Motif6	Motif7	Motif8	Motif9	Motif10
E-value	3.0 × 10^−2454^	3.6 × 10^−1957^	9.4 ×10^−1793^	1.2 × 10^−1544^	2.1 × 10^−1704^	1.3 × 10^−1271^	1.5 × 10^−1374^	1.5 × 10^−1722^	1.1 × 10^−996^	9.1 × 10^−638^
Sites	48	48	48	48	48	48	45	48	48	16
Width	50	41	41	34	41	30	41	50	29	44

## Data Availability

Not applicable.

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
