# Peer review of "Identification, Molecular Characterization, and Tissue Expression Profiles of Three Smad Genes from Water Buffalo (Bubalus bubalis)"

_genes, 2021, doi:10.3390/genes12101536_

Round 1

Reviewer 1 Report

Introduction
I am not convinced that to improve the water buffalo reproductive efficiency the molecular studies are needed. I would argue that your aim should be adjusted in order to support your actual analysis.

LINE 40 - "Gene breeding or genome technology..." - I do not understand this sentence.

Conclusions
I recommend to highlight who your findings will support water buffalo breeding world-wide

Figure 4 should be a heatmap to better present the results.
Figure 6 should be larger 

Reviewer 2 Report

Dear authors,

Line 1 – “Smads are involved in a variety of biological activities by mediating BMP signals.” – wrong sentence. No descriptions of used abbreviations.

Line 2 – Descript “CDSs” too.

Line 119 – add reference of using software

Line 120 – descript “AASs”

Line 140 – add country information of manufacturer of PCR mix.

Line 155 – isolated by RT-PCR – is it true?

Line 156 – How are you calculated length of amplicons in these gel-frames?

Line 165 – You are can write common color mark for all ABC parts

Line 190 – Text in figure too small. Shorter text mark (only species names) and increase it.

Line 207 – In figure all bootstraps are 0 according to legend. Is it true?

Regards,
